Stream baseline conditions shape functional responses to wastewater: evidence from insect-dominated sites

Enns Daniel daniel.enns@stud.uni-frankfurt.de 1 2
Baker Nathan Jay 3
Oehlmann Jörg 1 2
Jourdan Jonas 1 2
1 Department Aquatic Ecotoxicology, Johann Wolfgang Goethe Universität Frankfurt am Main , Frankfurt am Main , Germany
2 Kompetenzzentrum Wasser Hessen , Frankfurt am Main , Hessen , Germany
3 State Scientific Research Institute Nature Research Centre , Vilnius , Lithuania
Steinke Dirk
Electronic publication date: 2025 Oct 21
Publication date: 2025
Volume: 13
Electronic Location ID: e20193
Received 2025 Feb 12; Accepted 2025 Sep 15
Copyright: ©2025 Enns et al.
Copyright year: 2025
Copyright holder: Enns et al.
License: This is an open access article distributed under the terms of the Creative Commons Attribution License, which permits unrestricted use, distribution, reproduction and adaptation in any medium and for any purpose provided that it is properly attributed. For attribution, the original author(s), title, publication source (PeerJ) and either DOI or URL of the article must be cited.
License URL: https://creativecommons.org/licenses/by/4.0/

Keywords: Functional diversity, Spatial analysis, Lotic ecosystems, Ecological modeling, Macroinvertebrates, Organic pollution

Funding: The Kompetenzzentrum Wasser Hessen This work received funding from the Kompetenzzentrum Wasser Hessen. The funders had no role in study design, data collection and analysis, decision to publish, or preparation of the manuscript.

==============================
Wastewater treatment plants (WWTP) are a crucial part of modern day infrastructure, cleaning about half of our global wastewater. However, the emergence of micropollutants and higher frequencies of extreme weather events pose unprecedented challenges for society and biodiversity. Conventionally treated wastewater and altered flow regimes create environmental boundaries in rivers, impacting aquatic communities. Previous studies revealed pronounced taxonomic changes in freshwater invertebrate communities in response to WWTP effluents. To explore whether these shifts extend to functional diversity, we studied 338 communities upstream and downstream of 169 WWTPs using commonly applied functional diversity metrics. Surprisingly, we found no clear changes in functional alpha and beta diversity metrics, or community weighted means (CWM), suggesting that trait redundancy offsets the functional impact of the previously observed species turnover. However, in streams dominated by Ephemeroptera, Plecoptera and Trichoptera (EPT), we found more pronounced shifts in CWMs, indicating that the extent of functional changes depends on the baseline condition of the streams. EPT-dominated site-pairs showed significant shifts in traits related to reproduction, dispersal, and feeding, including increased occurrences of ovoviviparity and interstitial locomotion potentially as an avoidance mechanism of high flow and low oxygen saturation. Further, shifts in shredding and absorbing feeding types, aquatic passive dispersal, and hololimnic life cycles might be forms of adaptation to increased nutrient concentrations and reduced intermittency induced by WWTPs. These findings demonstrate that functional responses to wastewater inputs can remain undetected due to the noise inherent in large datasets and are often absent as a result of functional redundancy. In contrast, significant changes emerge in communities dominated by sensitive species, underscoring the value of trait-based approaches for detecting context-dependent ecological impacts.

Introduction

Rivers have historically been used as waste deposits (Lofrano & Brown, 2010) and, in contemporary society, remain intricately connected to sewage systems and their wastewater treatment plants (WWTP). Consequently, rivers commonly receive treated wastewater discharge, a significant achievement since untreated wastewater had previously been discharged into river ecosystems, causing deleterious and often unquantifiable consequences for biodiversity (Haase et al., 2023). Even today, only 52.8% of global wastewater is treated, though this figure is skewed toward developing countries which often lack large-scale treatment facilities (Jones et al., 2021). Conventional WWTPs treat wastewater in three steps: (1) removing suspended solids, (2) eliminating nutrients and organic compounds via biological degradation, and (3) removing phosphates. After treatment, the water is discharged back into rivers, where the treated effluent can sometimes account for up to 100% of the receiving river’s water volume (Hamdhani, Eppehimer & Bogan, 2020). Accordingly, the natural characteristics of rivers may be altered, and downstream regions affected. For example, despite treatment, effluent tends to be nutrient-rich and can have a different temperature regime (Kinouchi, Yagi & Miyamoto, 2007; Canobbio et al., 2009) compared to natural flows. Moreover, WWTPs are a major input pathway for emerging contaminants and micropollutants, since most treatment facilities lack the costly technology required for micropollutant removal (Margot et al., 2013; Münze et al., 2017; Moon et al., 2008). These micropollutants consist of a wide variety of chemicals such as pharmaceuticals, personal care products, hormones, and pesticides, which are especially detrimental for freshwater insects (Markert, Guhl & Feld, 2024b).

With effluent entering rivers, new environmental conditions can suddenly arise, with river sections below WWTPs generally showing unpredictable hydrographs with uncharacteristically high flow conditions during stormwater overflow (Canobbio et al., 2009; Markert, Guhl & Feld, 2024a). Introduced nutrients also promote the growth of periphytic algae and macrophytes (Murdock, Roelke & Gelwick, 2004), while fine sediments can congest the interstitial space (Sánchez-Morales, Sabater & Munoz, 2018). Furthermore, most WWTPs are not equipped to fully eliminate synthetic chemicals from effluents, thus impacted river sections can also exhibit altered chemical environments (Hörchner et al., 2024; Margot et al., 2013; Münze et al., 2017). These combined and interacting impacts can have multiple consequences for biodiversity, including shifts in fish community structure (Markert, Guhl & Feld, 2024a), reductions in invertebrate richness (Peschke et al., 2014) and diversity (Baker & Greenfield, 2019), changing invertebrate community structure (Enns et al., 2023), and the composition of invertebrate feeding groups (Ortiz & Puig, 2007).

Notably, Enns et al. (2023) showed the large-scale impact of 170 WWTPs on the taxonomic composition of macroinvertebrate communities, demonstrating that changes in macroinvertebrate community composition were driven by higher rates of taxonomic turnover (i.e., beta diversity), while alpha diversity metrics remained largely unaffected. Consequently, organisms unable to cope with the altered conditions created downstream of WWTPs can thus either be completely lost, displaced, and/or replaced by more tolerant species, causing community restructuring and the potential loss of ecosystem function.

In order to investigate such ecosystem functionalities, trait-based approaches have become established in recent years (Mammola et al., 2021; Bolnick et al., 2011; Jarzyna & Jetz, 2016; Menezes, Baird & Soares, 2010) which can more appropriately compare changes in biodiversity at various spatio-temporal scales (Múrria, Iturrarte & Gutiérrez-Cánovas, 2020). The traits of an organism are any features that potentially affect fitness and determine how species interact with one another, where a species can survive, and how much it contributes to ecosystem functioning (Cadotte, Carscadden & Mirotchnick, 2011). However, quantifying ecosystem functioning can be challenging since some traits are directly linked to key roles in the ecosystem, while others are indirectly linked. For example, the number of shredding organisms have a large impact on the breakdown of coarse particulate organic matter, while reproductive rates determine population size, thereby indirectly affecting litter breakdown. Indeed, through the lens of functional ecology and the available trait information (not considering hidden trait diversity; Cadotte, Davies & Peres-Neto, 2017), the sense of species converts into confined assortments of traits, weighted by abundance, and quantified by myriad functional diversity measures. Functional diversity is therefore a term used for a collective of metrics quantifying the richness, divergence, and regularity of traits on different levels of organisation, for example within and between individuals, populations, and communities (Mammola et al., 2021). Various metrics have been developed to quantify functional diversity and address fundamental ecological questions, such as changes in ecosystem processes (Dıáz & Cabido, 2001), species trait filtering (Pakeman, 2011), and ecosystem resilience (Rosenfeld, 2002). Unlike taxonomic approaches, which typically ignore disproportionate contribution of species to ecosystem functioning (Thompson & Starzomski, 2007; Jarzyna & Jetz, 2016; Cadotte, Carscadden & Mirotchnick, 2011), trait-based methods combined with monitoring data can effectively assess the impacts of invasive species (Guareschi et al., 2021), restoration measures (Coccia et al., 2021), and functional changes over large spatiotemporal gradients (Haase et al., 2023). An important concept in trait-based ecology is functional redundancy (Biggs et al., 2020) which describes the extent to which multiple species share similar trait combinations and ecological roles—meaning that the loss or addition of species does not always translate into functional change. Thus, trait-based community analyses provide new insights and fundamental knowledge that can aid practitioners in prioritising areas of importance for biodiversity conservation (Petchey & Gaston, 2002; Rosenfeld, 2002; Strecker et al., 2011).

Applying trait-based community analyses to extensive monitoring datasets, such as those from the Water Framework Directive (WFD), offers several advantages, including vast geographic applicability, effective stressor indication, as well as reliable and easily-implemented methodologies (Menezes, Baird & Soares, 2010). Yet, sub-setting such data by attributes of interest has the potential to increase the clarity of results but reduce the generality of findings (Catford et al., 2022). Accordingly, members of the Ephemeroptera, Plecoptera, and Trichoptera (hereafter EPT) fulfil a wide variety of functions within riverine ecosystems (Cibik et al., 2021; Schmidt-Kloiber & Hering, 2015), thereby being vital components to freshwater ecosystem functioning. These taxa are merolimnic (i.e., both aquatic and terrestrial life phases) and are known for their sensitivity to pollution. As a result, EPT taxa are commonly targeted in biological assessments and conservation efforts, are often used indicator taxa (Haase et al., 2023), and are analysed in studies relating functional diversity to WWTPs (Arce et al., 2014; González et al., 2023; Heß et al., 2024; Williams-Subiza et al., 2024).

While not commonly investigated, WWTP-induced shifts in ecosystem functioning have been observed for aquatic microbial (Ruprecht et al., 2021; Wakelin, Colloff & Kookana, 2008) and macroinvertebrate communities (Gücker, Brauns & Pusch, 2006; Burdon et al., 2023; Ortiz, Marti & Puig, 2005; Statzner et al., 2001; Mor et al., 2019; Péru & Dolédec, 2010), and have been based on a wide variety of functional endpoints. Here, we investigate changes in macroinvertebrate functional diversity between 338 communities located upstream and downstream of 169 WWTPs. We hypothesize that functional changes will be buffered and less pronounced compared to the previously identified taxonomic changes (Enns et al., 2023), as the replacement of taxa with functionally similar taxa provides a degree of resilience rooted in redundancy (Bêche & Statzner, 2009; Mori et al., 2015). Furthermore, we hypothesize that a more focused analysis of sites dominated by EPT taxa—indicative of lower impact and greater environmental sensitivity—will reveal more pronounced shifts in functional diversity metrics. By reducing the noise from the predominance of impacted and homogenized communities, this approach may better capture subtle functional responses.

Methods

Macroinvertebrate data and study design

Monitoring data used in this study were derived from the Hessian Agency for Nature Conservation, Environment, and Geology, covering the Rhine-Main metropolitan area in the south and rural areas in the north of Hessen, Germany (see Enns et al., 2023 for details). The larger river catchments covered in this study are the Rhine, the Main, as well as the Weser and their respective tributaries (map of study region in Enns et al. (2023); WWTP overview in Table S3). Monitoring data encompassed macroinvertebrate assemblage data, which were collected using standardized WFD methods (Haase et al., 2004) from 2006 to 2014, with additional samplings conducted in 2017 and 2018. The sampling season for all sites ranged from February to July. The taxonomy of the monitoring data follows the German operational taxon list (Haase, Sundermann & Schindehütte, 2006), which defines the minimum identification requirements for Germany under the WFD and includes classifications at varying taxonomic levels, primarily from family to species.

To assign monitoring sites to each WWTP, an upstream and downstream site for each WWTP was selected according to the criteria described in Enns et al. (2023). In total, 169 site pairs located above and below WWTPs were selected for our main dataset. It is important to note that 34 sites had an intermediate location between a possible upstream and a possible downstream site. These intermediate sampling sites were treated both as an upstream site of one pair and a downstream site of the other pair. This was necessary in order to not further limit the dataset. In cases where site pairs were sampled in multiple years only the most recent year was selected for the analysis. Within site pairs, the average time between sampling dates was 6.5 (± 19.3 SD) days. The minimum distance between a WWTP and the respective downstream sampling site was 70 m and the maximum distance was 27,356 m. WWTPs are characterised in Table 1, showing that most are grouped into size classes two and four, with average population equivalents of 2,765 and 34,754 human inhabitants respectively. Most of the analysed WWTPs employ a mechanical and biological treatment with nitrification, denitrification, and phosphate precipitation (Fig. S1). Regularly monitored effluent parameters were available and included biological oxygen demand, total ammonia, and total phosphorus concentrations, which tend to increase with decreasing size class, indicating that smaller WWTPs tend to emit higher organic pollutant loads in our dataset.

Table 1 Characterisation of WWTPs over all 169 sampling pairs.

Values are grouped by the WWTP size class (Table S2) and displayed as means ± standard deviations.

Size class	No. of WWTPs	Connected households	Human population equivalents	BOD
mg/L	NH4+
mg/L	Total P mg/L	
1	23	536 ± 263	590 ± 291	10.8 ± 11	7.1 ± 9.4	2.5 ± 2.4	
2	47	2,253 ± 931	2,779 ± 1,017	6.9 ± 5.1	4.8 ± 5.5	1.8 ± 0.9	
3	25	5,990 ± 2,346	7,615 ± 1,496	4.4 ± 1.8	1.3 ± 1.1	1.7 ± 1.0	
4	70	23,553 ± 13,322	34,455 ± 19,333	3.6 ± 1.7	1.1 ± 1.0	0.7 ± 0.4	
5	4	166,000 ± 73, 539	248,750 ± 85,865	3.8 ± 0.8	0.8 ± 0.6	0.4 ± 0.2	
Notes.

Abbreviations BOD biological oxygen demand

NH4+ ammonia

Total P total phosphorus

Values of BOD, NH4+ and Total P are based on effluent measurements.

Trait selection and extraction

In order to more accurately assign trait data to each taxon in our dataset, taxa were validated beforehand using the taxa validation tool from freshwaterecology.info (Schmidt-Kloiber & Hering, 2015). The trait data, extracted from freshwaterecology.info, comprised mostly traits derived from Tachet et al. (2010). For our analyses, nine trait groups representing 49 trait modalities were selected. We selected traits that were likely to be impacted by WWTPs through colmation of the interstitial, changes in flow and temperature regimes, and through the introduction of emerging contaminants, including: aquatic life stage, dispersion, feeding group, locomotion and substrate relation, reproduction cycles, reproduction modes, resistant forms, respiration, and fecundity (Arenas-Sánchez et al., 2021; Descloux, Datry & Usseglio-Polatera, 2014; White et al., 2017). While the traits are not explicitly designed to detect chemical pollutants, they represent broader ecological responses to the combined impacts of wastewater discharges. In cases where taxa had multiple aquatic life stages (i.e., mostly members of the Coleoptera), trait data were always assigned at the adult level. We used a trait gap filling approach to ensure good trait coverage across our taxonomic dataset (see Haase et al., 2023), with trait aggregation having been found to be a good proxy for expert trait allocation (Kunz et al., 2022). This resulted in a trait coverage of 100% across all taxa, with 32.4% of traits coded at the original taxon level, 1% at the sub-species level, 44.5% at the genus level, and 22.2% at the family level. Taxa with insufficient trait coverage were raised to the next highest taxon, for which sufficient coverage was available, resulting in 444 final taxa. Trait data were fuzzy coded and affinity scores were transformed into relative frequencies. From the taxa-trait matrix, a Gower dissimilarity distance matrix was constructed and subsequently used to perform a principal coordinate analysis (PcoA) in the ade4 R package (Thioulouse et al., 2018). To determine the occurrence probability of different trait combinations, we used kernel density estimations (Chacon & Duong, 2018), calculated on two principal components of the functional space which explained 25.6% of variance.

Community weighted means

Community-weighted means (CWM) are used to quantify the average value of a set of traits within a community, weighted by the abundance of each species. In other words, CWMs weigh each trait modality, of the relative frequency trait matrix, by the abundance of taxa assigned to that trait and averages them for each community. We used the FD R package (Laliberté, Legendre & Shipley, 2014) for the calculation of the CWMs. Delta CWM values, calculated as the upstream CWM minus the downstream CWM for each site pair, were tested against zero using one-sample Wilcoxon tests.

Alpha diversity indices

We calculated commonly used trait-based metrics measuring functional alpha diversity for all sites as well as functional beta diversity between corresponding site pairs. Due to the constraint of the number of taxa on functional components (Villéger, Mason & Mouillot, 2008), functional richness (FRic) was calculated using three functional components (m), explaining 36,1% of variance, while functional evenness (FEve) and functional dispersion (FDis) were calculated using the entire trait space. The calculations were performed using the FD R package following the code of Múrria, Iturrarte & Gutiérrez-Cánovas (2020). The optimal number of functional components was estimated by calculating the mean squared-deviation between the initial functional distance and the standardised distance in the functional space. FRic was calculated as the hypervolume of a convex hull in functional space (Villéger, Mason & Mouillot, 2008). FEve was calculated using the minimum spanning tree method (Villéger, Mason & Mouillot, 2008) and is a measure of how regularly taxa are distributed within the functional space. FDis was calculated as the mean distance of every taxon to the centroid of the community in functional space (Laliberté & Legendre, 2010). We further calculated functional redundancy as the quotient of Rao’s quadratic entropy divided by Simpson diversity of a community (Ricotta et al., 2016), using the adiv R package (Pavoine, 2020). Functional redundancy of upstream sites was tested against downstream sites via a paired Wilcoxon test.

Beta diversity indices

Functional beta components were calculated using the mFD R package (Magneville et al., 2022), whereby a different functional space was built and beta diversity was calculated using three functional dimensions explaining 28.2% of the variance. Functional beta diversity metrics measure the dissimilarity of occupied functional space between two or more communities. It can therefore be further decomposed into functional turnover, which indicates changes in functional composition, and a nestedness-resultant component, measuring to what degree communities resemble functional subsets of each other (Villéger, Grenouillet & Brosse, 2013). A high functional beta diversity can therefore either result from high turnover, meaning that communities are differently structured in regards of their functional strategies, or from a high nestedness-related component, indicating that functional strategies hosted by one community represent a small subset of another community. By deducing the intensity of the different components, one can infer if niche differentiation (in case of high turnover), niche filtering (in case of high nestedness), or functional convergence (in case of low functional beta diversity) drives the observed functional patterns between two or more communities (Villéger, Grenouillet & Brosse, 2013).

Null models

To account for the intrinsic relationship between taxonomic richness and functional diversity metrics (functional diversity is constrained by taxonomic diversity; Gotelli & Graves, 1996), we calculated null models by shuffling species names within the trait matrix through 999 randomisations (Swenson, 2014). This way, we kept the number of taxa and their abundances within each community constant, only changing their functional categorization. Trait associations were shuffled using the total taxa pool so as not to limit the number of possible randomisations to a point where statistical deductions became impossible. Calculation and analysis of null models for beta functional diversity metrics followed Villéger, Grenouillet & Brosse (2013). To be able to compare metrics between communities, standardised effect sizes (SES) were calculated by subtracting means of the null distribution from the observed values and dividing this value by the standard deviation of the null distribution (Múrria, Iturrarte & Gutiérrez-Cánovas, 2020; Baker et al., 2023). To determine if the observed SES values for each site (alpha diversity metrics) and site pair (beta diversity metrics) were outside the null distribution (a test of statistical significance and an indicator of a deterministic effect; Larsen et al., 2024), a two tailed test was used. Subsequently, we calculated the proportion of sites and site pairs with significantly different observed SES values to the null distribution. Differences in alpha diversity metrics, SES values between upstream and downstream sites, and between components of beta diversity SES were tested using a Wilcoxon test. Further, beta diversity SES were correlated with number of connected households, population equivalents, as well as log transformed biological oxygen demand, ammonia, and total phosphorus concentrations using a Spearman rank correlation coefficient test. These effluent parameters are regularly monitored by the relevant authorities and were therefore the only reliable and accessible indicators.

EPT dominant site analysis

Sites of good ecological status are prime targets for conservation practitioners and EPT taxa are generally viewed as sensitive towards pollution (Sinclair et al., 2024). To determine whether sites with a higher dominance of EPT organisms were more vulnerable to the impacts of WWTPs, we generated a subset of sites, keeping only those site pairs that had a proportion of at least 50% EPT individuals in the upstream community (hereafter EPT dominated sites). This resulted in 60 communities from 30 site pairs for which the same analysis as in ‘Trait selection and extraction’ was performed. The functional space was reconstructed based on these communities and functional alpha metrics were calculated using 8 dimensions, explaining 69.9% of variance and beta metrics were calculated using three dimensions, explaining 30.4% of variance. A brief taxonomic characterisation of subset of sites is given in (Section 2.1S).

Results & Discussion

Functional trait changes of the communities

Functional changes differed depending on whether we considered the complete dataset or the EPT dominated dataset. In the EPT dominated dataset, we found clear differences in the trait characteristics of macroinvertebrate communities up- and downstream of WWTPs (Fig. 1A), with differences becoming less evident when we considered all sampling sites together (Fig. 1B). At EPT dominated site-pairs, we found significant changes in trait combinations related to reproduction (voltinism, fecundity, reproduction technique), dispersal (aquatic life stages, dispersion technique, locomotion), and feeding (feeding type), with all these traits significantly deviating from zero (Fig. 1A).

Figure 1 Changes of trait community weighted means between sampling sites upstream and downstream of WWTPs.

Delta community weighted means of traits for the EPT dominant subset (A) and the complete dataset (B). Shifts to the right indicate an increase in CWMs below WWTPs, while shifts to the left indicate a corresponding decrease. Asterisks represent significant shifts from zero (one-sample Wilcoxon test; p-value < 0.05).

Traits with a positive shift, meaning an increase in the richness and/or abundance of species with these traits, were ovoviviparity, interstitial locomotion, shredding and absorbing feeding types, aquatic passive dispersal, aquatic adults (i.e., hololimnic life cycle), and low fecundity (<100). The observed shifts in trait diversity, both positive and negative, indicate which traits are being filtered by WWTPs and point toward communities composed of more generalist organisms and fewer EPT taxa. Organisms with the aforementioned trait combinations often gain competitive advantages under certain stress conditions, compared to those that are less adaptable or more sensitive (Paz et al., 2022). For example, ovoviviparous organisms carry their eggs and potentially regulate certain environmental conditions, e.g., avoidance behaviour or oxygen regulation through gill fanning. Further, interstitial locomotion might be more beneficial compared to epibenthic modes of locomotion like crawling or swimming, since the organism is able to hide under gravel during high flow events, e.g., storm overflows of WWTPs (Marino et al., 2024). In addition, organisms with absorption feeding habits might profit from the increased nutrient loads typical of wastewater effluents.

Traits with a negative shift between up- and downstream sites were univoltinism, aquatic egg stage, eggs as resistant forms, cemented and isolated eggs, high fecundity (between 1,000 and 3,000), aerial active dispersion, and scraping feeding types, with the latter five showing generally high affinities in Ephemeroptera and Plecoptera, but also some Trichoptera, Coleoptera and Gastropoda (Fig. S2). In particular, the Ephemeroptera and Plecoptera are sensitive toward trace metals and municipal wastewater (Let et al., 2022), and thus WWTPs may be influencing environmental filters which select against these traits. For example, insects with aerial active dispersion often emerge during the late evening or at night and suffer from light pollution (Perkin et al., 2014). It is thus reasonable to assume that WWTPs do not impact this trait directly, yet their spatial proximity to settlements and the resulting increased light pollution does. Further, nutrients introduced by effluents would promote the growth of epibenthic algae, thus favouring organisms with scraping feeding types. However, since this trait is associated with many ephemeropteran, trichopteran, and coleopteran taxa, the feeding type appears to be only passively selected against by WWTPs. Péru & Dolédec (2010) found that, in terms of biodiversity and functional metrics, river sections downstream of WWTPs deviate stronger from minimally disturbed reference conditions compared to sites located upstream of WWTPs. Similar to our results, traits with the strongest contribution to the deviation were related to reproduction types as well as locomotion and substrate relation, with the former potentially being attributed as a coping response against effluents and their impacts (e.g., organic pollutants, sedimentation, etc.). Furthermore, Mor et al. (2019) showed that traits such as small body sizes, semivoltine life cycles, eggs as aquatic stage, eggs as resistance forms, and gill respiration decreased downstream of WWTPs. This partially coincides with our results, with the only difference being that, in our results, the trait ‘gill respiration’ is not changing and semivoltine lifecycle is increasing, though this deviation may have resulted from differences in analytical methodologies. It could be expected that an increase in shredders and a decrease of scraping feeding types results in increased litter break down rates and a stronger biofilm formation. While Pereda et al. (2020) report these expected patterns, Burdon et al. (2023) and Englert et al. (2013) observed decreased invertebrate driven decomposition, with a comprehensive meta-analysis by Brauns et al. (2022) also showing lower leaf litter decomposition and a higher net ecosystem production in wastewater impacted streams. Nevertheless, the changes in CWMs observed in our study suggest that sites with a higher proportion of EPT taxa, and by extension their unique functional roles, are altered by downstream WWTPs, corroborating a previous study by Enns et al. (2023).

When considering the entire dataset of all 169 site pairs, trait compositional shifts downstream of WWTPs were much less apparent (Fig. 1B). Here, only traits related to fecundity and locomotion significantly deviated from zero. However, the absolute difference in average abundance between upstream and downstream sites was generally higher when we only considered the EPT dominant subset (absolute difference in average abundance for taxa with prevalence higher than 50% in EPT dominated subset: 16.8; in complete dataset: 3.7). This could explain the lower deviation of CWMs, since they are influenced more by highly abundant taxa (Péru & Dolédec, 2010). Another factor is the increased heterogeneity of communities of the complete dataset, which contributes to the indistinctness of the CWM response. This suggests that WWTPs tend to have a more substantial impact on the functional diversity of communities where insect taxa comprise a majority and further shows how results of an upstream-downstream comparison can differ depending on the baseline (upstream) community condition. In general, interpretations of changes in CWMs are not always straightforward, since stressors can select directly or indirectly for or against traits. This circumstance is detailed in Verberk, van Noordwijk & Hildrew (2013), whereby traits are connected to each other by the species that are associated with them. Thus, without a good understanding of species autecology and accompanying environmental and habitat data, it is difficult to disentangle the mechanisms behind how WWTPs filter traits and by extension the species to which they are associated.

Changes measured on alpha diversity scale

In both datasets, no functional alpha diversity indices showed a significant difference between sites up- and downstream of WWTPs (Fig. S4). Functional redundancy is often regarded as a valuable metric for conservation, since it gives information on the vulnerability of functional strategies (Mouillot et al., 2014; Ricotta et al., 2016). Thus, stable functional redundancy between site pairs can also be viewed positively. Contrarily, Péru & Dolédec (2010) as well as Mor et al. (2019) reported lower functional diversity below WWTPs, the latter showing that the impact magnitude depends on streambed substrate.

Changes measured on beta diversity scale

Most site pairs showed no significant deviations from null model expectations, suggesting niche convergence rather than differentiation. In other words, the set of functions fulfilled by communities up- and downstream of the WWTPs are, for most site pairs, quite similar. This finding was consistent when we considered the EPT dominant sites separately and when we repeated our analyses using complete data from all 169 site pairs. This therefore suggests that WWTPs are generally not changing the composition of the functional strategies of invertebrate communities we analysed. However, when considering all 169 site pairs, 14% of the site pairs (23 pairs) deviated significantly from the null model expectations, while only 10% of site pairs (three pairs) deviated significantly in the EPT-dominated site subset. Further, standardised effect sizes of functional beta components showed no significant correlation with WWTP characteristics from Table 1 (Fig. S3). Thus, it is reasonable to assume that in most cases (86%), the impact of WWTPs does not lead to detectable changes in the functional strategies of macroinvertebrate communities. However, this should not be interpreted as evidence that WWTPs are incapable of driving functional change. Rather, such effects may remain undetected due to current data limitations—particularly the lack of traits directly linked to chemical sensitivity—and the potential buffering capacity of functional redundancy. The low observed functional beta diversity hints at regional environmental pressures, as opposed to local factors like WWTPs, homogenising functional strategies across assemblages. Evidence for strong local factors influencing the functional composition of communities comes from intermittent rivers, where intermittency is a strong driver of functional turnover in invertebrate communities (Aspin et al., 2018; Piano et al., 2020; Viza et al., 2024). There is no evidence to date that chemical stressors—even when present in low concentrations—cause similarly strong effects on community functionality (Alric, Geffard & Chaumot, 2022). Thus, for intermittent rivers, WWTPs can even have a stabilizing effect on functionality since they guarantee a more regular water supply (Brooks, Riley & Taylor, 2006).

Functional diversity: limitations, interpretation, and future considerations

Functional diversity measures are a valuable tool to infer and understand important ecological circumstances. However, two major shortcomings exist thus far. First, if the environment selects for species with certain traits, other traits indirectly get selected for because they are intrinsically associated to that species (and not the other way around). Second, complex environmental pressures might select for a certain combination of traits, rather than a single trait. Thus, trait patterns are context dependent and result in low discriminatory power of diversity metrics, potentially hindering the interpretation of causalities between stressors and observed trait patterns (Heino, Schmera & Eros, 2013; Verberk, van Noordwijk & Hildrew, 2013). For our study area and period, access to comprehensive environmental data was limited—particularly with regard to measurements of chemical loadings in both the effluent and receiving waters. In addition, the other potentially confounding factors, such as changes in the surrounding land-use practices (Vandewalle et al., 2010) and the condition of the in-stream and riparian habitats (Calapez et al., 2021; Sargac et al., 2021), could not be assessed. Nevertheless, these problems could be mitigated by an a priori selection of relevant traits and/or by standardizing relevant environmental factors across study sites (Statzner & Beche, 2010). The former is a relatively easy task, achievable by reviewing existing literature. However, data on environmental factors are often either missing and/or spatiotemporally incongruent with the invertebrate sampling sites (Santos et al., 2021).

While functional diversity indices can give insights into the status of traits and functional strategies of communities, from which theoretical conclusions can be drawn, the relationship between indices and in-field functional processes remains unclear. There are several factors constraining the relationship between ecosystem functions and functional diversity. First, trait data are inherently generalized and often aggregated at higher taxonomic levels, which can obscure species-specific and intraspecific variability (Bolnick et al., 2011). Moreover, trait expression is influenced by complex, context-dependent interactions with environmental conditions, including adaptive responses that may alter trait expression in response to local stressors (Jourdan et al., 2024; Jourdan et al., 2019); (Hernández-Carrasco et al., 2025). Second, the relationship between functional diversity and ecosystem processes is constrained by the non-independence of traits and abiotic factors (Hamilton et al., 2020). Litter decomposition, as an example, is a highly relevant process in freshwater ecosystems and has a direct link to food and feeding traits of invertebrates. Studies suggest that organic matter breakdown in streams is either not correlated with invertebrate functional diversity indices and breakdown related traits (Vos & Schäfer, 2017), or it is negatively related with functional richness (Rideout et al., 2022), showing that we are far from a complete understanding of the relationships between ecosystem processes and functional diversity. Further in-field measurements of community composition and ecosystem processes, together with mesocosm experiments, would provide valuable data, which, through new analytical tools, could bridge the gap between theory and practice for many freshwater ecosystem functions.

Conclusions

In our urbanized world, wastewater treatment is crucial, but without costly upgrades to expand capacity and add quaternary treatment steps, WWTPs can adversely affect river environments and biological communities and their associated ecosystem functions. While we found that changes in functional diversity between sites up- and downstream of WWTP are less pronounced than taxonomic diversity (Enns et al., 2023), thereby supporting our hypothesis, we also note that changes in functional diversity are context dependent. A broad, holistic analysis may obscure functional changes, but focusing on specific subsets—such as sites dominated by EPT taxa—reveals more pronounced shifts. Though this targeted approach limits the generality of our conclusions to these particular communities, it remains highly relevant for conservation efforts and environmental management, as EPT-dominated sites often hold significant ecological value. The reality is that although WWTPs do not appear to have a substantial impact on local biodiversity, site pairs with higher proportions of sensitive EPT insect taxa tend to have more impoverished trait combinations downstream of WWTPs, showing more substantiated changes in functional diversity. This poses a threat to the metacommunity structure as these sites can function as donor populations for neighbouring sites once WWTP capacity has been increased and modernizations installed. We issue caution in directly implicating WWTPs as the sole drivers of the observed patterns, with many other confounding factors also potentially playing a role. However, given the spatial coverage of our dataset and its comprehensiveness (a benefit of having more data), we can conservatively conclude that WWTPs do instil some sort of pressure on freshwater communities, revealed as changes in trait CWMs, albeit with controlling mechanisms still evading our complete understanding. We suggest that coupled environmental data be the bare minimum for environmental agencies, as this will allow the more direct assessment of WWTPs on freshwater ecosystems. Further, careful site and trait selection has the ability to more accurately determine the stressors of freshwater systems. Accordingly, the results of such functional inclusions can inform practitioners about stressors related to ecosystem functions on a wide geographical range.

Supplemental Information

Supplemental Information 1 Supplemental Materials

We would like to thank the Hessian Agency for Nature Conservation, Environment and Geology, in particular Thomas Wanke and Björn Michaelis, for providing the monitoring data.

Additional Information and Declarations

Competing Interests

Author Contributions

Data Availability

Jörg Oehlmann is an Academic Editor for PeerJ.

Daniel Enns conceived and designed the experiments, performed the experiments, analyzed the data, prepared figures and/or tables, authored or reviewed drafts of the article, and approved the final draft.

Nathan Jay Baker conceived and designed the experiments, analyzed the data, authored or reviewed drafts of the article, and approved the final draft.

Jörg Oehlmann conceived and designed the experiments, authored or reviewed drafts of the article, and approved the final draft.

Jonas Jourdan conceived and designed the experiments, performed the experiments, authored or reviewed drafts of the article, and approved the final draft.

The following information was supplied regarding data availability:

The code and supplementary data are available at Zenodo: Daniel Enns. (2025). ChickentartR/Functional-ecosystem-dynamics-downstream-of-wastewater-treatment-plants: Stream baseline conditions shape functional responses to wastewater (v1.0.0). Zenodo. https://doi.org/10.5281/zenodo.16745291.

The macroinvertebrate monitoring and Wastewater treatment plant effluent data from the Hessian Agency for Nature Conservation, Environment and Geology can be requested from Dr. Thomas Wanke at (department email: kontakt@hlnug.hessen.de): https://www.hlnug.de/themen/wasser/fliessgewaesser/fliessgewaesser-biologie/ueberwachungsergebnisse/fischnaehrtiere.

The wastewater treatment plant effluent data can be requested from Dr. Björn Michaelis at (department email: kontakt@hlnug.hessen.de): https://www.hlnug.de/themen/wasser/abwasser/kommunales-abwasser-in-hessen/betrachtung-hessischer-kommunaler-klaeranlagen-zur-steigerung-der-ammoniumelimination

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
