# Peer review of "Stream baseline conditions shape functional responses to wastewater: evidence from insect-dominated sites"

_PeerJ, doi:10.7717/peerj.20193_

## Round 0.1 · original submission · Major Revisions

Both reviewer provided very detailed and important comments and made suggestions on how to address their concerns. Please ensure to respond to all the comments provided. Especially, concerns related to data integration and statistical analyses should be addressed carefully.

Reviewer 1 ·

Basic reporting

Considerations and Comments
The study applies a solid approach to evaluating macroinvertebrate communities but lacks a clear connection with WWTPs and the environmental characteristics of the sites. To strengthen this relationship, it would be essential to include information on the exact distance of sampling points from WWTPs, water quality data from the effluents, and hydrological characteristics of the river, such as flow rate and seasonality.

To better contextualize the results, it is recommended to include a map showing the location of the WWTPs and the upstream and downstream sampling station pairs, as well as explicitly reporting the minimum and maximum distances between sampling points and the WWTPs.

Although the number of WWTPs included, the study period, and the study region are mentioned, it is not clearly stated which specific rivers were evaluated or which WWTPs were analyzed. This limits the ability to interpret the effects based on the local conditions of each river system.

Regarding functional diversity indices, there is insufficient information on their values and variability between sampling sites. Additionally, the taxonomic level used for the identification of macroinvertebrates is not clearly specified, and a detailed list of species and their average abundances at each site is not provided. Since functional analyses can be affected by taxonomic resolution, it is recommended to explicitly state the level of identification used and assess its impact on the results.

Regarding the quality of the discharged water, the study only reports mean values for BOD, NH₄, and Total P, without justifying why only these parameters were considered and not others. If data on other contaminants (e.g., heavy metals, conductivity, micropollutants) are available, it is recommended to include them and evaluate their influence on the biological community. Conductivity, for example, is an easily measurable parameter and can significantly affect macroinvertebrates, especially in WWTPs without advanced treatment.

The characterization of WWTPs is based solely on the equivalent population served, without including information on the type of treatment used. It is suggested to add a table in the supplementary materials detailing the type of treatment for each WWTP (conventional, nutrient removal, advanced, etc.), as this may influence the detected impacts.

To improve the interpretation of the effects of WWTPs on communities, it is recommended to:

• Include scatter plots or violin plots showing the concentrations of water quality parameters in the effluent during the study period.
• Incorporate a gradient analysis to evaluate how biological changes vary as a function of distance from the WWTP, rather than just comparing upstream and downstream sites.
• Analyze hydrological data, such as river flow, flow velocity, and precipitation, as these factors determine the river’s capacity to dilute contaminants and may influence the impacts of WWTPs.
• Evaluate the contaminant load upstream of the river, as the biological community’s response may depend on whether the ecosystem was already impacted before the discharge.
• Group and analyze effects by river, considering not only the proximity to the WWTP but also the treatment efficiency and contaminant load of each system.
• Use statistical approaches such as Linear Mixed-Effects Models (LMER) to account for hierarchical data structures, considering site pairs as random effects to better isolate the impact of WWTPs.
• Consider applying Structural Equation Modeling (SEM) to assess direct and indirect relationships between WWTP characteristics, water quality parameters, and macroinvertebrate community changes, allowing for a more comprehensive understanding of causal mechanisms.

Overall, the study provides a robust analysis of the macroinvertebrate community structure, but its ability to attribute biological changes to WWTPs is limited by the lack of environmental and spatial data. Incorporating these elements would significantly improve the validity and applicability of the findings.

Experimental design

The study employs a robust approach to evaluating macroinvertebrate communities; however, its ability to directly link observed biological changes to WWTPs is limited by certain aspects of the experimental design. The lack of on-site water quality measurements (e.g., BOD, ammonia, phosphorus, conductivity, heavy metals, or emerging contaminants) makes it difficult to determine whether the observed community shifts are directly attributable to WWTP effluents or other environmental factors. Additionally, hydrological factors such as river flow, seasonal variation, and dilution capacity were not considered, despite their crucial role in modulating effluent impacts. The study reports only an average distance between sampling sites and WWTPs without analyzing whether community responses vary with proximity to the discharge. Furthermore, the taxonomic resolution used for macroinvertebrate identification is unclear, which could influence the accuracy of functional trait analyses. Lastly, the characterization of WWTPs is limited to population equivalent, without providing details on treatment type, efficiency, or specific pollutant loads. These factors limit the study’s capacity to establish a well-supported causal relationship between WWTPs and biological changes in aquatic communities.

Validity of the findings

The study provides valuable insights into the functional trait changes in macroinvertebrate communities, but the validity of its findings is constrained by the lack of key environmental data and spatial considerations. The absence of direct water quality measurements at sampling sites makes it difficult to confirm whether the observed biological changes are directly caused by WWTP effluents or by other environmental factors. Additionally, the study does not account for hydrological variables such as river discharge, seasonal variation, and dilution capacity, which are critical for assessing the extent of effluent influence. The lack of a distance-gradient analysis further limits the ability to evaluate how the impact of WWTPs varies with proximity to the discharge. Furthermore, no significant correlations were found between WWTP characteristics and functional diversity changes, which raises questions about whether WWTPs are the primary driver of the observed patterns or if other unmeasured factors are contributing. Without addressing these aspects, the study’s conclusions about the effects of WWTPs on macroinvertebrate communities remain suggestive rather than definitive.

Additional comments

This study provides a valuable contribution to understanding the functional responses of macroinvertebrate communities in the context of WWTPs. The use of trait-based approaches is a strength, offering ecological insights beyond traditional taxonomic assessments. The dataset is extensive, covering multiple WWTPs and site pairs, which adds robustness to the analysis. Additionally, the focus on EPT-dominated sites highlights the sensitivity of key aquatic taxa to environmental disturbances.

An important aspect to consider is the limited availability of certain environmental data across all sites, which is a common challenge in large-scale ecological studies. However, while it may not always be feasible to obtain a complete set of hydrological, water quality, and river discharge data for every location, it is essential to make full use of the data that are available. For example, the study reports average distances between sampling sites and WWTPs, but does not present individual site distances or the exact locations of WWTPs. Providing such information would enhance spatial analyses and allow for a clearer interpretation of how proximity to effluents influences macroinvertebrate communities.

Balancing broad-scale synthesis with detailed case studies is key to maximizing the study’s impact. While summarizing trends across multiple WWTPs is valuable for generalization, presenting more detailed analyses for sites where environmental data are available would strengthen the depth of interpretation.

Overall, this research addresses an important environmental issue and has the potential to inform future assessments of WWTP impacts. With some refinements in data integration and statistical approaches, it could serve as a strong reference for functional ecology studies in freshwater ecosystems.

Reviewer 2 ·

Basic reporting

The manuscript titled "Functional ecosystem dynamics downstream of wastewater treatment plants" (#113824) proposes a timely topic for better understanding the impacts of wastewater treatment plant (WWTP) effluents on riverine communities. The study applies common approaches in studying the functional diversity of benthic macroinvertebrate communities. Negative or insignificant change patterns should be considered as valuable as positive results; However, my main concern is the limited findings from this study, which may also be influenced by the nature of the data or study approaches. Given these limitations, it does not seem justified to develop separate papers on taxonomic shifts (Enns et al., 2023) and functional diversity dynamics (this paper), especially when the key finding is a shift in macroinvertebrate communities towards more generalist organisms and fewer EPT taxa, without significant changes in common functional diversity metrics.
Below, I provide some suggestions for improved clarity in the study approaches focusing on functional diversity shifts associated with WWTP effluents:
1. Title: The reference to functional ecosystem "dynamics" in the title is misleading, as no clear changes were found in the functional diversity metrics of the whole community. Consider clarifying the "dynamics" aspect by specifying whether it pertains to the whole community or sub-communities, or revise the title to better reflect the study’s findings.
2. Abstract:
• L14-18: The benefits of trait-based approaches are well established in the literature and may not need to be highlighted as a key takeaway from this study. Additionally, this study does not seem to demonstrate added value in using functional traits, as the observed responses were driven by dominant EPT taxa composition rather than the functional diversity employed. Could it be that the common functional diversity metrics used in this study are not sensitive indicators of WWTP effluents?
• L9, 13: Can the authors specify the particular traits and directions of functional shifts in benthic macroinvertebrates in response to WWTPs (e.g., details from L245-255)? Providing these specifics would offer more useful insights for readers than the general statement that functional traits are changing or shifting.
• L16-18: If using the entire dataset introduced noise and prevented clear conclusions regarding WWTP impacts, wouldn’t it be more informative to analyze functional diversity separately for rare/sensitive insects versus dominant/insensitive taxa sub-communities and incorporate statistical analyses to link these biotic patterns to potential drivers linked to WWTP sources?

Experimental design

3. Introduction:
• Species group selection (L95-100): It remains unclear why the authors chose macroinvertebrate to investigate the effects of WWTPs. The fact that this group is not commonly studied in this context may indicate a lack of mechanistic relevance. Provide supporting evidence to justify this choice, for example, by including information and references on ecosystem function shifts mentioned in L95-98.
• Study hypotheses (L99-107): The first hypothesis suggests that functional changes are less pronounced than taxonomic changes, which appears contradictory to the rationale for using trait-based approaches. Clarify how this hypothesis aligns with the study’s objective of assessing functional diversity shifts.
4. Methods:
• Study sites (L115-121): Were these site selection criteria already established in a previous study? If so, this section could be shortened by referring to that study. Alternatively, provide more details on the characteristics of the study sites in this study to help clarify their relevance to functional shifts in macroinvertebrate communities.
• Site selection (L125-126): Treating pairs of sites sampled in different years as independent sites introduces pseudo-replication, violating the assumption of statistical independence. Observations from the same sites, even across different years, are likely correlated. Additionally, this approach leads to a loss of potential insights into temporal dynamics in species responses. The reviewer suggests excluding the limited multi-year dataset from the analysis of spatial dynamics in species' functional diversity.
• Table 1: Are these the average values of variables for each site pair? I would expect information on the difference in water quality within each sampling pair—i.e., the delta changes between upstream and downstream WWTP sites—since these changes are associated with WWTP effluents and may influence the functional beta diversity of benthic macroinvertebrates (e.g., L330-331, or Figure 1). Additionally, the table presents pollutant concentrations in effluents (expressed in mg/L), which differs from pollutant loads (i.e., concentration multiplied by flow, typically expressed as kg/day or tons/year, depending on the scale of analysis).
• L141-144: None of the selected traits (aquatic life stage, dispersion, feeding group, locomotion and substrate relation, reproduction cycles, reproduction modes, resistant forms, respiration, and fecundity) have a direct association with chemical pollution (e.g., pesticides or pharmaceuticals) or with elevated flow or temperature from WWTP effluents. Please provide supporting evidence justifying their relevance, or consider redefining the functional trait metrics to better capture the potential impacts of WWTPs.
• L172-175: The choice of functional components (m) is influenced by community size—i.e., in more diverse communities with higher taxonomic richness, a low m may fail to capture community patterns. Was m set to three for extremely low-diversity communities with only five taxa, or was this value applied across all communities? For larger communities, a functional dimension of six (Mouillot et al., 2021) to eight (Arias-Real et al., 2021) may still be insufficient to accurately represent the functional space.
Mouillot, D., Loiseau, N., Grenié, M., Algar, A.C., Allegra, M., Cadotte, M.W., et al., (2021). The dimensionality and structure of species trait spaces. Ecology Letters, 24: 1988-2009. https://doi.org/10.1111/ele.13778.
Arias-Real, R., Gutiérrez-Cánovas, C., Menéndez, M., Granados, V., Muñoz, I. (2021). Diversity mediates the responses of invertebrate density to duration and frequency of rivers' annual drying regime. Oikos, 130: 2148-2160. https://doi.org/10.1111/oik.08718.
• L148-150: Only one-third of taxa have original trait values, while two-thirds rely on aggregated trait values at the genus and family levels. Could this lack of detailed trait data contribute to the weaker functional diversity responses observed via CWMs compared to the taxonomic composition turnover? The first hypothesis—that increased functional redundancy is present—might instead be an artifact of the limited availability of trait information for the studied taxa. Would the results differ if the analysis were limited to sites and communities where at least 60-70% of original taxa had complete trait data?
• L215-218: Establishing causal linkages with environmental indicators of WWTPs should be central to investigating the drivers of upstream-downstream macroinvertebrate community dynamics. It is therefore unjustifiable that the authors rely solely on Spearman correlation tests, as correlation does not imply causation. Furthermore, these correlations are only linked to beta diversity SES instead of other functional metrics across the 182 site pairs.

Validity of the findings

5. Results and Discussion:
• Sections 3.1 to 3.3: The shift in functional diversity, whether for the whole community or EPT taxa, should be discussed in relation to changes in abiotic metrics of WWTP effluents, particularly when abiotic data is available. Without linking these shifts to actual changes in water quality upstream and downstream of WWTPs—which vary by site depending on pollutant intensity, toxicity, and the nature of chemical mixtures—the explanation for functional diversity changes appears unconvincing (e.g., L325-327).
• L331-333: Stating that the impacts of WWTPs are not strong enough to cause changes in functional strategies is difficult to accept, given the selected functional metrics and the nature of the unmeasured trait data used in this study. This is especially true considering that a previous study demonstrated a clear taxonomic shift in species composition driven by WWTP effluents. Did the authors attempt to run scenarios with increased pollutant concentrations from WWTP effluents to determine the threshold at which specific chemicals exert a ‘strong enough’ effect?
• L358-360: Is this limitation relevant to the dataset used in this study? Which missing environmental data and variables are being referred to?

Additional comments

6. Others:
• L18: remove the redundant words ‘rare species’ in the phrase ‘rare species communities dominated by sensitive insect orders’.
• L48-57: Specify and discuss some impacts that altered chemical environments have on biodiversity rather than listing the references. A single study by Enns et al. (2023) is not enough when the paper is focusing on impacts of WWTPs on biodiversity.
• L59: This study investigates the sub-community (EPT taxa) to community-level variations, which does not include the individuals within species. Thus, it is questionable why the authors introduce the irrelevant term inter- and intra-specific variation, which appeared only once in the Introduction.
• L128: Which ‘population' is being referred to—human or species? Additionally, what is the unit of measurement for this variable?

---

## Round 0.2 · accepted · Accept

All reviewer comments were sufficiently addressed by either modifying parts of the manuscript accordingly or providing context as to why authors disagree with reviewers. This assessment has been confirmed by one of the reviewers who read the updated version. I think the manuscript is now acceptable for publication.

Reviewer 2 ·

Basic reporting

no comment

Experimental design

no comment

Validity of the findings

no comment

Additional comments

The authors have satisfactorily addressed point-by-point comments from the reviewer.